# New Preclinical Development of a c-Met Inhibitor and Its Combined Anti-Tumor Effect in c-Met-Amplified NSCLC

**DOI:** 10.3390/pharmaceutics12020121

**Published:** 2020-02-03

**Authors:** Nam Ah Kim, Sungyoul Hong, Ki Hyun Kim, Du Hyung Choi, Joo Seok Kim, Kyung Eui Park, Jun Young Choi, Young Kee Shin, Seong Hoon Jeong

**Affiliations:** 1College of Pharmacy, Dongguk University-Seoul, Gyeonggi 10326, Korea; namah87@dongguk.edu (N.A.K.); kihyun@dongguk.edu (K.H.K.); 2Research Institute of Pharmaceutical Sciences, College of Pharmacy, Seoul National University, Seoul 08826, Korea; sungyoul@snu.ac.kr; 3Department of Pharmaceutical Engineering, Inje University, Gyeongnam 50834, Korea; choidh@inje.ac.kr; 4Department of Pharmacy, College of Pharmacy, Seoul National University, Seoul 08826, Korea; ansrkd5@snu.ac.kr; 5R&D Center, ABION Inc., Seoul 08394, Korea; pku1218@abionbio.com (K.E.P.); comfly@abionbio.com (J.Y.C.); 6Department of Molecular Medicine and Biopharmaceutical Sciences, Seoul National University, Seoul 08826, Korea

**Keywords:** c-Met tyrosine kinase inhibitor, PDX, bioavailability, poorly water-soluble, NSCLC

## Abstract

c-Met is a receptor tyrosine kinase with no commercially available product despite being a pivotal target in cancer progression. Unlike other c-Met inhibitors that fail clinically, ABN401 is a newly synthesized c-Met inhibitor that is not potentially degraded by aldehyde oxidase (AO) in human liver cytosol. This study aimed to determine the physicochemical stability, pharmacokinetics in beagle dogs, and therapeutic effect of ABN401 in a c-Met-amplified non-small-cell lung cancer (NSCLC) patient-derived xenograft (PDX) model. ABN401 was found to be a weak basic compound, with pKa and log *P* values of 7.49 and 2.46, respectively. It is poorly water-soluble but soluble at acidic pH. The accelerated storage stability is dependent on temperature, but the purity remains at over 97% after 6 months. The bioavailability is approximately 30% in dogs and it is highly efficient in the PDX model, achieving around 90% tumor growth inhibition in combination with erlotinib. These observations indicate that the compound is acceptable for the next phase of trials.

## 1. Introduction

c-Met is a receptor tyrosine kinase for hepatocyte growth factor (HGF) and plays a critical role in embryonic development, tissue repair, and tumor progression [1,2]. The mutation, overexpression, and amplification of c-Met are often observed in cancers such as gastric cancer and non-small-cell lung cancer (NSCLC) at different stages and even after treatment with epidermal growth factor receptor (EGFR) tyrosine kinase inhibitors (TKIs) [2]. At present, alternative treatments such as immunotherapy have have been developed, giving new hope to many sufferers of the diseases, but chemotherapy still remains the ‘standard’ therapy beyond progression [3]. Approaches after the emergence of TKIs resistance are still required for better therapies.

The first generation of EGFR TKIs such as gefitinib and erlotinib, approved in 2002 and 2004, respectively, is widely used for the treatment of patients with NSCLCs. These therapies are effective against tumors that express active mutant forms of EGFR such as del19 and L858R. Mutant forms are reported to be expressed in about 35% of NSCLCs with aggressive tumor growth [4,5]. Nevertheless, EGFR TKIs exhibit great initial efficacy after administration but the NSCLCs inevitably acquire resistance within 1 or 2 years [3]. In addition, Met amplification is reported to occur in almost 21% of patients with NSCLCs after treatment with the first generation of EGFR TKIs and in about 3%–7% of patients without the treatment [6,7]. This suggests that combined treatment with EGFR TKIs and c-Met inhibitors may be necessary in a subset of patients to circumvent the onset of resistance. However, further generations of c-Met inhibitors are still under investigation, with on-going clinical trials of 15 compounds [8,9]. Although c-Met has been targeted by several drug discovery programs, the studies have resulted in many failures, delaying market release. The first generation of c-Met (i.e., ATP-competitive) inhibitors lacked target selectivity and inhibited other targets such as vascular endothelial growth factor and Axl receptor tyrosine kinase by XL880 and BMS-777607, respectively [10,11]. The second generation (Figure 1a) had relatively better selectivity to c-Met, with a similar structure, especially the presence of a quinoline group, which seems to be crucial for selective binding. However, research on JNJ-38877605 and SGX523 was terminated during clinical trials due to renal toxicity [12,13,14,15]. It was concluded that quinoline-containing chemical structures can be metabolized in a species-specific manner to form poorly soluble nephrotoxic metabolites, explaining the observed renal toxicity of SGX523 and JNJ38877605 [13,14,16]. The major metabolites of SGX523 and JNJ38877605 have been confirmed as 2-quinolinone-SGX523 and M3 (i.e., quinoline group), respectively [13,14]. Therefore, the second-generation structure of c-Met inhibitors exhibited selectivity but was liable to toxicity.

ABN401 is structurally differentiated from known inhibitors. It is a triazolopyridazine, a small molecule that is effective in inhibiting c-Met activity [8,17]. In this study, its physicochemical evaluation was performed in terms of pH solubility, log *P*, forced degradation, storage stability, oral gavage formulation development, and pharmacokinetic profile in beagle dogs. In addition, the study included an investigation of the efficacy at high potency and the selectivity of ABN401 based on histopathologic and genetic analyses of PDX models for NSCLC.

## 2. Materials and Methods

### 2.1. Materials

ABN401, with a molecular weight of 566.66 g/mol (C_29_H_34_N_12_O), was developed and supplied by ABION Inc. R&D center (Seoul, Korea). The chemical structure of ABN401 is illustrated in Figure 1a. Using an online server (http://admet.scbdd.com), values for polar surface area (PSA), number of hydrogen bond donors, number of hydrogen bond acceptors, and rotatable bond count of ABN401 were calculated as 119.04, 0, 13, and 7, respectively. The compound was stored in a freezer (<0 °C) as received. All other reagents were analytical or HPLC grade and used as received.

### 2.2. Metabolic Stability against Aldehyde Oxidase

ABN401 (1 µM in 0.3% DMSO) was incubated with human liver cytosol (BioIVT, NY, USA) in the absence and presence of an aldehyde oxidase inhibitor. Human liver cytosol (0.5 mg/mL) and 0.1 M phosphate buffer (with 0.1 mM EDTA) at pH 7.4 were pre-incubated at 37 °C prior to the addition of ABN401. The final incubation volume was 500 μL. Phthalazine, which is known to be metabolized by AO, was used as a control compound. Incubations were also performed in the presence of the AO inhibitor, menadione, at 100 μM. The reaction times were set to 0, 5, 15, 30, 60, and 120 min. The reactions were stopped by transferring an aliquot of the incubate to an organic solvent containing the internal standard at the appropriate time points. The termination plates were centrifuged at 2500 rpm for 30 min at 4 °C to precipitate the protein. Following protein precipitation, the sample supernatants were combined in cassettes of up to four compounds and analyzed using liquid chromatography-tandem mass spectrometry (LC-MS/MS). From a plot of the integration peak area ratio (compound peak/internal standard peak area) against time, the gradient of the line was determined. Subsequently, the half-life and intrinsic clearance were calculated using the equations below:Elimination rate constant (k) = (−gradient)(1)
Half-life (t_1/2_) (min) = 0.693/k(2)
Intrinsic clearance (CL_int_) (μL/min/mg protein) = (V × 0.693)/t_1/2_(3)
where V = Incubation volume (μL)/Protein (mg). The percentage of ABN401 remaining at each time point in the absence and presence of the inhibitor, along with the intrinsic clearance value (CL_int_), half-life, and standard error of the CL_int_ were recorded.

### 2.3. Forced Degradation Study

ABN401 was dissolved in 50/50 (v/v) % of acetonitrile and water, to give a stock solution of 2 mg/mL. The stock sample was diluted to a final concentration of 0.5 mg/mL and subjected to various stresses such as 1.0 N HCl acid (pH 1.1), 1.0 N NaOH base (pH 12.3), 1.0% hydrogen peroxide, heat (in solution and as a solid material) in an oven at 60 °C, and UV-Vis light (in solution) in a closed quartz container at 20 °C with UV light at 16.08 mW/m2; Vis 5.7 kLux. The samples were tested with quantitative HPLC method at 24, 48, and 72 h. The results were expressed in area % for recovery. The control sample was left at ambient conditions to simulate the potential loss of ABN401 when left at ambient conditions (> 99.0%).

### 2.4. log P, log D, and pKa Calculations

Solutions at pH values of 1 to 12 were prepared using HCl and NaOH. At first, 10^−1^ N, 10^−2^ N, 10^−3^ N, 10^−4^ N, 10^−5^ N, and 10^−6^ N of HCl and NaOH solutions were prepared and their pH values were measured with a pH meter (Metrohm 827 pH lab, Zurich, Switzerland). One milliliter of the 10^−4^ N, 10^−5^ N, and 10^−6^ N of HCl and NaOH solutions was withdrawn and diluted in a 100 mL flask for more coordination. In each prepared solution (*n* = 3), excess ABN401 was dissolved and mixed for 24 h. All samples were centrifuged for 1 min at 12,000 rpm. The pH solubility of ABN401 was analyzed by HPLC and the pKa was determined by plotting a graph of log(solubility) against pH.

ABN401 solution (10 μg/mL) was mixed with octanol at a ratio of 1:1 in a glass vial and placed horizontally in a Digital Bio Rotator (SeoLin Bioscience, Seongnam, Korea), and shaken at a constant 90 rpm for 24 h. After 6 h of equilibrium under room temperature, two phases of water and octanol were separated using a separating funnel. The experiment was repeated five times. The concentration of ABN401 at each phase was analyzed by HPLC and log *P* and log D were calculated using the Henderson-Hasselbalch equation below with the previously determined pKa value.
(4)$$\log D_{bases}\;=\log P+\log⎣\frac{1}{\left(1+\;10^{pK_{a}-pH}\right)}⎦$$

### 2.5. Differential Scanning Calorimetry (DSC)

Thermograms for ABN401 were obtained using a Q-2000 DSC (TA Instruments, New Castle, DE, USA). Each sample (i.e., around 3 mg) was sealed in an aluminum pan and a blank pan was used as a reference. Both the reference and sample pans were first kept at 20 °C for 5 min to ensure isothermal starting conditions. DSC analysis was performed at a scanning rate of 10 °C/min from 20 to 180 °C under a nitrogen flow of 50 mL/min. Onset and melting temperatures (*T_m_*) were evaluated using the Universal analysis 2000 software provided with the equipment.

### 2.6. Thermogravimetric Analysis (TGA)

TGA was performed to observe changes in the weight of ABN401 using a thermogravimetric analysis system (TGA Q50, TA Instruments, New Castle, DE, USA). The sample (around 10 mg) was placed on a sample pan and heated at a rate of 10 °C/min from room temperature to 250 °C. The result was evaluated using the Universal analysis 2000 software.

### 2.7. Dynamic vapor sorption (DVS)

Moisture sorption analyses were utilized using a VTI-SA vapor sorption analyzer (TA Instruments, New Castle, DE, USA). API was placed into the sample holder. The percentage of relative humidity (% RH) in the chamber was increased from 5% to 95% RH for moisture adsorption, followed by a decrease in % RH from 95% to 5% for moisture desorption. A drying phase (i.e., 40 °C, less than 0.01% weight loss in 2 min) was set to run before to the analysis. The temperature was set at a 25 °C throughout the analysis. The program criteria were set to 0.0001% weight change or 2 min stability of weight gained or lost before the program could proceed to the next set parameter.

### 2.8. Powder X-Ray Diffraction (PXRD)

PXRD patterns were measured at room temperature using a D8 Advance with a DAVINCI X-ray diffractometer (Bruker AXS GmbH, Karlsruhe, Germany) equipped with a Ni-filtered Cu-Kα radiation (λ = 1.54056 Å) and a high speed LynxEye detector. The powder samples were loaded onto a quartz holder and scanned over a range of 4–40° at a scanning rate of 6°/min.

### 2.9. Granulation

High shear wet granulation was carried out using a Mi-Pro (ProCepT, Zelzate, Belgium) with a capacity of 0.9 L Prior to the granulation, the physical mixture was passed through a sieve with a mesh size of 30, transferred to a granulation bowl, and then premixed for 5 min at the impeller speed of 600 rpm. This was followed by the addition of water using a peristaltic pump. The liquid-to-solid ratio was 60% (*w*/*w*, water was added to powder mixture). This value was set based on the preliminary study. The batch size for the preparation was 4 g of ABN401 and the wet massing time was set to 3 min. The wet granules were dried in an oven at 45 °C for 5 h and formulated into size 00 gelatin capsules at 200 mg and size 01 capsules at 50 mg.

### 2.10. Scanning Electron Microscope (SEM)

A scanning electron microscope (Model EM-30, COXEM, Daejeon, Korea) was used to characterize the morphology of the granules. The samples were coated with gold under vacuum in an argon atmosphere prior to observation. The SEM images of the samples were taken at an acceleration voltage of 20 kV.

### 2.11. Dissolution Profile

A dissolution test was carried out according to USP 36 Apparatus 2 (paddle method) at 100 rpm using 900 mL of dissolution medium maintained at 37 ± 0.5 °C (Varian 705 DS; Varian, Cary, NC, USA). Each capsule was enclosed in a stationary sinker (20 × 12 mm) to prevent the capsule from floating on the surface of the dissolution medium or sticking to the inner surface of the dissolution vessels (*n* = 4). The drug content was analyzed with an HPLC system using samples collected in Eppendorf tubes and centrifuged for 1 min.

### 2.12. Pharmacokinetic Studies

Male beagle dogs were purchased from Marshall, Beijing, China. They were housed under controlled humidity, temperature, and a 12L:12D lighting schedule. Before the experiment, the animals were fed less than 300 g/day for 2 days and those with a body weight around 7.0 ± 0.5 kg were selected for the pharmacokinetic study (*n* = 3). They were fasted overnight before the experiments but allowed to drink water. This study was reviewed, assessed, and approved by the Institutional Animal Care and Use Committee (IACUC) of the Korea Institute of Toxicology (KIT). The project identification codes were N117008 and N116047, signed on 31 May 2017. For the experiments, 0.5-mL blood samples were collected from each animal at selected intervals and loaded into K2-EDTA tubes. After centrifugation at 13,200 rpm for 5 min, quantitative analysis was performed using LC-MS/MS. Acetic acid buffer (0.1 M) at pH 4.0 with 20% PEG400 was selected as the intravenous solution for ABN401 and WinNonlinTM (Version 5.2.1, Pharsight, USA) was used to calculate the pharmacokinetic parameters (i.e., T_1/2_: terminal half-life, C_max_: maximum observed peak concentration, T_max_: time to reach C_max_, AUC_last_: area under the time-concentration curve from zero to the last quantifiable time-point, AUC_inf_: area under the time-concentration curve from zero to infinity, MRT_last_: mean residence time from zero to the last quantifiable time-point). The bioavailability (BA) of each dose was calculated using the following equation:Bioavailability = (AUC_last_po_/AUC_last_iv_) × 100(5)

### 2.13. High Performance Liquid Chromatography (HPLC)

An HPLC system (Shimadzu LC-20, Shimadzu, Kyoto, Japan) was used to analyze the solubility, dissolution profile, and degradation products as well as to determine the storage stability of ABN401. The wavelength of the UV detector was set at 282 nm. For quantification analysis, the Agilent Eclipse Plus C18 (5 μm, 4.6 × 150 mm) (Agilent technologies, Santa Clara, CA, USA) was used and maintained at 30 °C. The mobile phase was a mixture of acetonitrile and 50 mM acetate buffer at pH 5.0 at a volume ratio of 50 to 50 (*v*/*v* %). The flow rate of the mobile phase was 0.5 mL/min and the injection volume was 10 μL. A Kromasil C8 (5 μm, 4.6 × 250 mm) column was used for qualification analysis to observe the impurity profile during the storage stability test. The following gradient was applied: 0–20 min, 80% acetonitrile and 20% acetate buffer to 20% acetonitrile and 80% acetate buffer, maintained up to 25 min; 25– 28 min, back to 80% acetonitrile and 20% acetate buffer, maintained up to 35 min. The flow rate of the mobile phase was 1 mL/min and the injection volume was 10 μL.

### 2.14. LC-MS

To evaluate the molecular weight of ABN401 and its degradation products, LC-MS analysis was conducted using an LCMS 2020 system (Shimadzu, Kyoto, Japan) with two LC-20AD pumps, CTO-20A column oven, SIL-20A autosampler, CBM-20A controller, SPD-20A DAD, and LCMS 2020 single quadruple mass spectrometer. The UV detection wavelength, column, temperature of the column oven, mobile phase composition, and flow rate were the same as the values used for the quantification HPLC method. Mass analysis was conducted using an electrospray ionization (ESI) source that produces ions in the positive ionization mode. The LC-MS spectra were acquired from m/z 50 to 800. The following parameters were used: nebulizing gas flow, 1.5 L/min; drying gas flow, 15 L/min; ESI interface temperature, 350 °C; desolvation line (DL) temperature, 250 °C; heat block temperature, 200 °C; ESI interface voltage, 4.5 kV; and detector voltage, 1.2 kV. Tuning of the mass spectrometer was performed using the auto-tuning function of LabSolutions LC-MS software.

### 2.15. LC-MS/MS

HPLC system (Agilent 1200, Santa Clara, CA, USA), consisting of a binary pump, a degasser, and a refrigerated autosampler (at 10 °C), was used in this study. Analytical samples were separated on the Agilent Eclipse XDB-C18 (3.5 um, 2.1 × 50 mm) at 23 °C. The mobile phase was composed of solvent A (10 mM ammonium formate in purified water) and solvent B (0.1 % formic acid in acetonitrile) and delivered at a flow rate of 0.3 mL/min. In this study, the gradient separation (i.e., the initial condition of the mobile phase consisted of 80% of solvent A and 20% of solvent B, then solvent A was decreased to 0% within 2 min and maintained for 5 min, followed by an increase to 80% within 0.2 min, maintained until 10 min) was used with a 10-min run time per sample. The MS system was composed of an API 4000 Sciex triple quadruple mass spectrometer (AB Sciex, Framingham, MA, USA). The triple quadrupole mass spectrometer was equipped with a turbo ion spray source. The MS/MS detection involved multiple reaction-monitoring in the positive ionization mode. The intensive tracer ion mass for each compound was 567.5 → 467.2 for ABN401 and 237.3 → 194.11 for carbamazepine as an internal standard. The conditions for the electrospray interface were: curtain gas, 20 psi; collision gas, medium; ion spray voltage, 5000 V; temperature, 500°C; ion source gas 1, 50 psi and ion source gas 2, 55 psi. The optimized voltage parameters for the analytes were as follows: declustering potential (61 V), entrance potential (10 V), collision energy (28 V), and collision cell exit potential (12 V). Data acquisition, quantification, and calculations were performed with the Analyst software version 1.4.2 (AB Sciex, Framingham, MA, USA).

### 2.16. Patient-Derived Xenograft (PDX) Studies

For efficacy studies with the patient-derived tumor xenograft mouse model, LU0858 human primary NSCLCs were established by implanting fresh surgical tumor tissues into immunodeficient mice. Tumors from stock mice bearing human primary NSCLCs were harvested, dissected into fragments, and inoculated into female BALB/c-nude mice. Each mouse was implanted subcutaneously at the right flank with a tumor fragment (P3–5, 2–4 mm in diameter) for tumor development. When the average tumor size reached approximately 150 mm^3^, the mice were randomly allocated to three groups with eight mice per group and treated daily with orally administered vehicle, 10 or 30 mg/kg of ABN401, and/or 30 mg/kg of erlotinib (Selleckchem Houston, TX, USA). ABN401 formulated in 20% PEG400 (Sigma-Aldrich, Saint Louis, MO, USA) in 0.1 M acetate buffer at pH 4.0 was administered every day for 3 weeks. All the procedures related to animal handling, care, and treatment in this study were performed according to guidelines approved by the Institutional Animal Care and Use Committee of CrownBio following the guidance of the Association for Assessment and Accreditation of Laboratory Animal Care (AAALAC). The project identification code was E1197-U1601, signed on 4 November 2016. Animals were housed in pathogen-free animal housing in individually ventilated cages (IVC) with access to sterilized food and water ad libitum. The animals were maintained on a 12 h light/dark cycle in a temperature- and humidity-controlled animal research facility. All efforts were made to minimize the suffering of the animals. All mice were observed daily and weighed during the treatment period. Tumor measurements consisted of both tumor-to-control and tumor growth inhibition (TGI) taken as endpoints to determine whether or not the tumor growth was delayed or the mice were cured. The tumor size was measured twice weekly in two dimensions using a caliper and the volume was expressed in mm^3^.
Volume (mm^3^) = length (mm) × width (mm)^2^ × 0.5(6)

## 3. Results

### 3.1. Metabolic Stability against AO in Human Cytosol

The clinical failure of c-Met inhibitors may be the result of insoluble metabolites generated by AO in the kidneys. ABN401 was incubated in the presence of human liver cytosol to determine the AO metabolic stability in the absence (-) and presence (+) of menadione, an AO inhibitor. The results indicated that the metabolite of ABN401 in human liver cytosol was very low since the clearance rate was not affected by the presence of menadione (Figure 1b). In comparison, phthalazine, which can be metabolized by AO, exhibited a very high clearance rate in human liver cytosol and was inhibited by the presence of menadione. The results suggested that AO activity potentially does not contribute to the metabolism of ABN401. Therefore, ABN401 has the potential to be developed as a new c-Met inhibitor.

### 3.2. Forced Degradation and Accelerated Storage Stability

Forced degradation of ABN401 under various environmental stresses (i.e., pH, heat, light, and oxidation) was performed in order to understand its physicochemical stability. The result is presented in Figure 1c. ABN401 was relatively stable in acidic and basic pH, which resulted in a difference of approximately 1.7% after 72 h of exposure. In addition, degradation was more favorable in the aqueous state than in the solid state and could be accelerated by heat. On the other hand, ABN401 in solution was highly sensitive to UV/Vis light followed by oxidation. Figure 2a presents the exact molecular weight of ABN401 and the effect of hydrogen peroxide exposure, which caused chemical degradation. Based on the results, ABN401 was stored away from any light source and air by placing it in a closed container (i.e., high-density polyethylene, HDPE) throughout the study.

After understanding the degradative factors for ABN401 (i.e., light and oxidation), accelerated storage stability is further evaluated for 6 months at different temperatures (i.e., less than 0 °C, 4 °C, 25 °C, and 40 °C; Figure 3) in a closed system. Seven peaks including the main peak and impurities were evaluated using qualitative HPLC method (Figure 2b). After 6 months of storage, ABN401 at 0 °C (i.e., frozen) and 4 °C exhibited a decrease in the main peak (i.e., < 0.5%) and increases in peaks 2 and 3, but almost no change in peaks 1, 4, 6, and 7 (Figure 3). A consistent decrease in the main peak (i.e., > 0.5%) was observed at 25 °C, whereas higher elevation was observed in peaks 2 and 3. In addition, peak 1 increased from 0.08% to 0.19%. These changes in peaks 1, 2, and 3 were more distinguishable when the storage temperature increased to 40 °C. The results suggest that peaks 1, 2, and 3 are from the degradants of ABN401 and temperature is an accelerating factor for the reaction. On the other hand, the total loss of ABN401 in all four conditions was only around 1.2%–3.1%.

### 3.3. Physicochemical Properties and Solubility Enhancement

Physicochemical properties can influence key product attributes including bioavailability, stability, and manufacturability. Moreover, the properties of drug candidates play an important role in understanding the compound’s overall behavior in vitro and in vivo. Parameters including log *P* (partition coefficient), log *D* (distribution coefficient), equilibrium solubility, and pK_a_ are typical intrinsic factors necessary for its product development. Firstly, the pH solubility of ABN401 was investigated at a pH range from 2.28 to 12.36 to calculate the apparent pK_a_ using validated quantitative HPLC method (limit of quantitation; 0.64 μg/mL) (Figure 2c). The solubility of ABN401 was relatively high in acidic conditions but was poor and nearly constant in basic conditions (Figure 1d). The highest solubility for ABN401 was measured at pH 2.28 (i.e., 26.87 mg/mL) and was equal to or less than 0.01 mg/mL above neutral pH. The results showed that ABN401 is a weakly basic compound and its tentative pK_a_ was found to be 7.49 via simple simulation. Secondly, log *D* was calculated by determining the concentration of ABN401 in organic and aqueous phases to calculate log *P*. Log *D* was directly calculated from five different runs, assuming that ABN401 is in ionic form in solution. After assigning log *D* and pK_a_ to the Henderson–Hasselbalch equation, the average log *P* for ABN401 was 2.46 ± 0.04 (Table 1).

The solubility of ABN401 in two different acidic buffers (i.e., acetate and citrate) were analyzed at three different pH values: 4.0, 4.5, and 5.0 (Table 2). In 0.1 M acetate buffer at pH 4.0, ABN401 exhibited relatively high solubility, whereas citrate buffer did not increase the solubility. Different concentrations of PEG400, cremophor EL, and polysorbate 80 were also evaluated as co-solvents to ABN401 in 0.1 M acetate buffer at pH 4.0 to investigate the solubility enhancement. PEG400 was the most effective of the three excipients. Therefore, 20% PEG400 in 0.1 M sodium acetate was selected as an oral gavage formulation and intravenous solution for in vivo studies. A short storage stability test was performed for 8 days in a cold chamber to determine the concentration recovery yields at low and high concentrations, which were 0.03 mg/mL and 133 mg/mL, respectively. In addition, three different pH values (i.e., 2, 3, and 4) were compared since the compound exhibited higher solubility at lower pH. As a result, pH 4.0 yielded the highest percentage recovery in both low and high concentrations (Table 3).

### 3.4. Thermal Properties and Stability on Crystallinity

ABN401 was thermally scanned with DSC and TGA to understand its thermodynamic properties by means of heat absorption and mass change, respectively. Figure 4a shows the thermogram for ABN401 with two endothermic peaks. The first endotherm is broad at temperatures less than 100 °C, followed by a second sharp endotherm, indicating *T_m_*. The heat of fusion was around 46.82 J/g and changed to 47.32 J/g with *T_m_* after drying in an oven at 40 °C for a month in an open container (Figure 4b). After storage, the weight loss decreased from 2.98% to 1.47% of the depth of the first endothermic peak. This phenomenon suggests that the first endothermic peak is affected by the solvent or water content in the solid. Figure 4c shows newly synthesized ABN401 with enhanced quality at a production scale that eventually showed less weight loss (0.99%) and higher crystallinity (i.e., higher *T_m_* value). On the other hand, DVS results showed typical DVS results for a pharmaceutical ingredient with the same weight before and after analysis, indicating ABN401 was relatively less adsorptive to moisture (i.e., > 65 % RH) (Figure 4d).

The polymorphism of ABN401 was also investigated with physical stresses by heat and humidity. Figure 4e presents the PXRD results for ABN401, which exhibited no change in its polymorph when exposed to 40 °C/75% RH for a month in an open container as well as for 3 months in a closed container. These results confirm the consistent crystallinity status of ABN401 during accelerated storage, including thermal analysis.

### 3.5. Dissolution Profile and Bioavailability in Beagle Dog 

For the drug product development, fifty milligrams of ABN401 was blended with pharmaceutical excipients at a ratio of 1:2 and filled into gelatin hard capsules for a dissolution study. Figure 5a shows the in vitro dissolution profiles of each prepared capsule with and without excipients in pH 1.2 medium. All samples that exhibited immediate release profiles above 85% dissolved in less than 45 min, as the solubility of ABN401 is higher in acidic environments. The fastest dissolution profiles were obtained for capsules with microcrystalline cellulose (MCC) followed by capsules formulated with starch, as these components may help disperse the compound in the release medium. Higher standard deviation was observed before 15 min due to variations in the time required for capsule opening.

For pharmacokinetic studies, ABN401 in capsules at three different doses (i.e., 1, 5, and 10 mg/kg) with MCC was administered to male beagle dogs to evaluate the pharmacokinetic profiles with various dose strengths (Table 4). A capsule without MCC (5 mg/kg) was also included in the study for comparison. The C_max_ and AUC increased with the dose from 1 to 5 mg/kg, but not at 10 mg/kg, and T_max_ ranged from 1.3 to 2.0 h across the doses. The C_max_ and AUC values showed a dose dependency at 1 and 5 mg/kg (i.e., approximately five-fold), but not at 10 mg/kg, suggesting poor absorption or saturation. Similarly, 5 mg/kg had higher bioavailability (i.e., 30.7%) with and without MCC. The standard deviation was lower for ABN401 mixed with MCC compared to the capsule without MCC, suggesting consistency with the excipient.

On the other hand, Figure 5b presents the dissolution profiles for two different doses (i.e., 50 and 200 mg) prepared via high shear granulation. The lower dose had slightly faster dissolution, but both doses reached 100% release within 60 min. For efficient product preparation and quality control in the clinical study, the excipients and granulation process were optimized to obtain a uniform blend (Figure 5d). Both 50 and 200 mg of the blend filled into capsules reached 100% release within 30 min in pH 1.2 medium (Figure 4c). However, the dissolution profile of ABN401 is influenced by the pH and the differences between the two doses increased because the solubility decreases as the pH increases.

### 3.6. In Vivo Efficacy in c-Met-Amplified NSCLC PDX Models

To investigate the efficacy of ABN401, c-MET-amplified NSCLC PDX models were utilized in the study. Tumors from mice bearing human primary NSCLC tumors were harvested, dissected into fragments (2–4 mm in diameter) and inoculated subcutaneously in the right flanks of BALB/c-nude mice. Mice were treated orally with 10 or 30 mg/kg doses of ABN401 when the average tumor volume reached 150 mm^3^. ABN401 and/or erlotinib were administrated once daily for 3 weeks. As shown in Figure 6, in the high c-Met-amplified LU0858 model with Met copy number >14, ABN401 at 10 or 30 mg/kg as a single treatment demonstrated dose-related anti-tumor activity, with tumor growth inhibition (TGI) values of 68.6% and 84.9%, respectively. The differences in anti-tumor activity were statistically significant when compared with the vehicle control (*P* < 0.001). Although the LU0858 PDX models had EGFR L858R mutation, erlotinib as a single-agent treatment at 30 mg/kg did not exhibit any anti-tumor activity, with no statistically significant difference compared with the vehicle control (*P* > 0.05). The combination of ABN401 with erlotinib showed anti-tumor activity, with a TGI value of 90.1%, which had a statistically significant difference compared with the vehicle control (*P* < 0.001). An additional anti-tumor effect was observed with combination therapy compared to ABN401 or erlotinib alone. These results suggest that ABN401 can significantly reduce tumor growth in the PDX model with c-Met-amplified NSCLC. Therefore, ABN401 showed potent anti-tumor activity against c-Met in vivo. Moreover, the Met copy number and protein expression level affected the effectiveness of the c-Met inhibitor.

## 4. Discussion

c-Met is one of the receptor tyrosine kinases and its aberrant activation plays an important role in the development of lung cancer [18]. Despite its importance, certain c-Met inhibitors have failed in clinical applications since the earlier candidates were metabolized by AO, causing renal toxicity. The AO metabolism has several points that make it difficult to analyze in preclinical phase. First, standard metabolic stability studies using liver microsomes may be inaccurate since AO are mainly present in the cytosolic fraction. Second point would be variability of AO activities during storage conditions. Third, in vivo studies on AO-mediated metabolism in animal models are problematic because of differences in the profound species [16,19,20]. Therefore, AO metabolism was only performed in human liver cytosol in the absence and presence of the AO inhibitor and a newly synthesized c-Met inhibitor, ABN401, was potentially not a substrate of AO.

Based on physicochemical experiments, ABN401 was found to be a weak basic compound, with poor solubility and pH dependency. Its log *P* was found to 2.46 ± 0.04, suggesting drug-like properties for oral administration. Forced degradation studies showed that light and oxidation were degradative stressors for the compound. Three relevant impurities (peaks 1, 2, and 3) increased after 6 months of storage and the peak amplitudes depended on the storage temperature while the main peak remained around 98% in all conditions. In addition, the degree of polymorphism did not change during the stability test in an open bottle with high humidity. Consequently, the optimal storage condition was determined to be under refrigeration to avoid an increase in impurities. The increase in impurity is expected to be less after controlling the contents of the solvent. Likewise, the storage stability of the filled capsules (i.e., granules and from new batch) differed by temperature but lower in value. Peak 1 and peak 2 increased when the capsules were stored at 40 °C, whereas relatively less change was observed when they were stored in a refrigerator for 6 months.

Fast dissolution was observed at pH 1.2 with and without bulking agents. In order to correlate with in vivo, a pharmacokinetic study was performed in beagle dogs. The T_max_ of orally administered ABN401 was attained within the first 2 h and the bioavailability was approximately 30% (supportively, the absolute bioavailability in rats was 45% [21]), except at 10 mg/kg (Table 4). The C_max_ for 10 mg/kg was similar to that for 5 mg/kg, suggesting low absorption from the gastrointestinal tract. This phenomenon may be due to a decrease in solubility from the pH change at higher concentrations (Figure 5c). Evidently, the difference in dissolution of two doses was highest at pH 4. The gastric pH in dogs is around 1.8 whereas the intestinal pH is around 7.3, suggesting a vigorous change in pH after administration [22]. In other words, supersaturation in solution can occur, leading to crystallization and resulting in a decrease in absorption. In summary, ABN401 as a physical blend exhibited fast absorption, dose-dependent C_max_, and moderate bioavailability but strong sensitivity to pH change in the gastrointestinal condition. Based on the results, high shear granulation process was adopted for faster dissolution and higher consistency in quality control for drug product for next phase.

In this in vivo study, we presented a highly potent inhibitor of c-Met and its dose-responsive anti-tumor efficacy in tumors with c-MET-amplified NSCLC. In this study, LU0858 was utilized as an NSCLC PDX model. Detailed array comparative genomic hybridization and next-generation sequencing analyses confirmed that this PDX model has a high copy number gain for the region containing c-MET with a copy number >14 as well as EGFR L858R mutation, which is one of most common mutations in NSCLCs. Erlotinib was the first small molecule inhibitor to be approved for NSCLCs and is generally referred to as a first-generation EGFR TKI [23]. Although tumors are sensitive to this agent, patients inevitably develop acquired resistance [24]. Thus, it is necessary to find better therapies, such as c-Met and/or Her-2 inhibitors, for patients with acquired resistance after treatment with first-generation EGFR TKIs. Our results in this study showed that erlotinib as a single agent treatment has no anti-tumor activity in the LU0858 model with EGFR-mutation NSCLC. The LU0858 model appears to correspond to patients with acquired resistance to EGFR TKIs. Interestingly, ABN401 did show effective anti-tumor activity in this model. In addition, additional anti-tumor effects were observed with combination therapy compared to ABN401 or erlotinib alone. Therefore, ABN401, as a highly potent and selective c-Met inhibitor, effectively suppresses the growth of c-Met-amplified NSCLC cells associated with EGFR mutation. 

## 5. Conclusions

ABN401, a new c-Met inhibitor, was potentially not a substrate to aldehyde oxidase, which is the root cause of clinical failures. An oral formulation was developed based on physicochemical and biopharmaceutical properties and administered to an NSCLC PDX model, achieving around 90% tumor growth inhibition in combination with erlotinib. ABN401 is currently being tested in a phase I/II trial in Austria (NCT04052971) and received Investigational New Drug (IND) approval in June 2019 in South Korea.

## Figures and Tables

**Figure 1 pharmaceutics-12-00121-f001:**
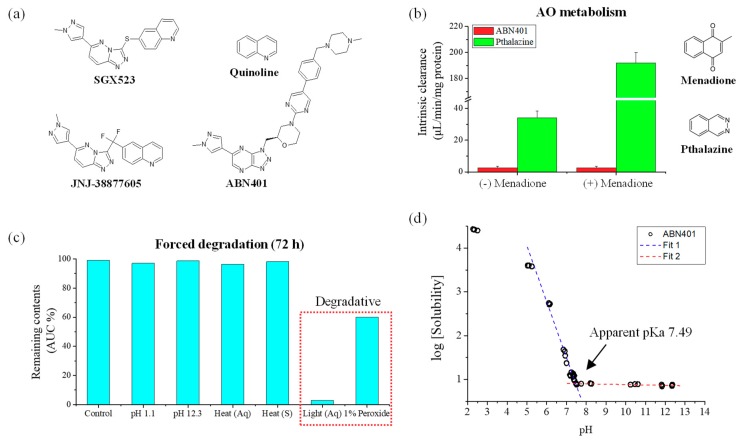
(**a**) Chemical structures of ABN401 and reference compounds. (**b**) Cytosol metabolic stability of ABN401 versus phthalazine. (**c**) Forced degradation result after exposure to different physical stresses. (**d**) pH solubility profile at various pH values.

**Figure 2 pharmaceutics-12-00121-f002:**
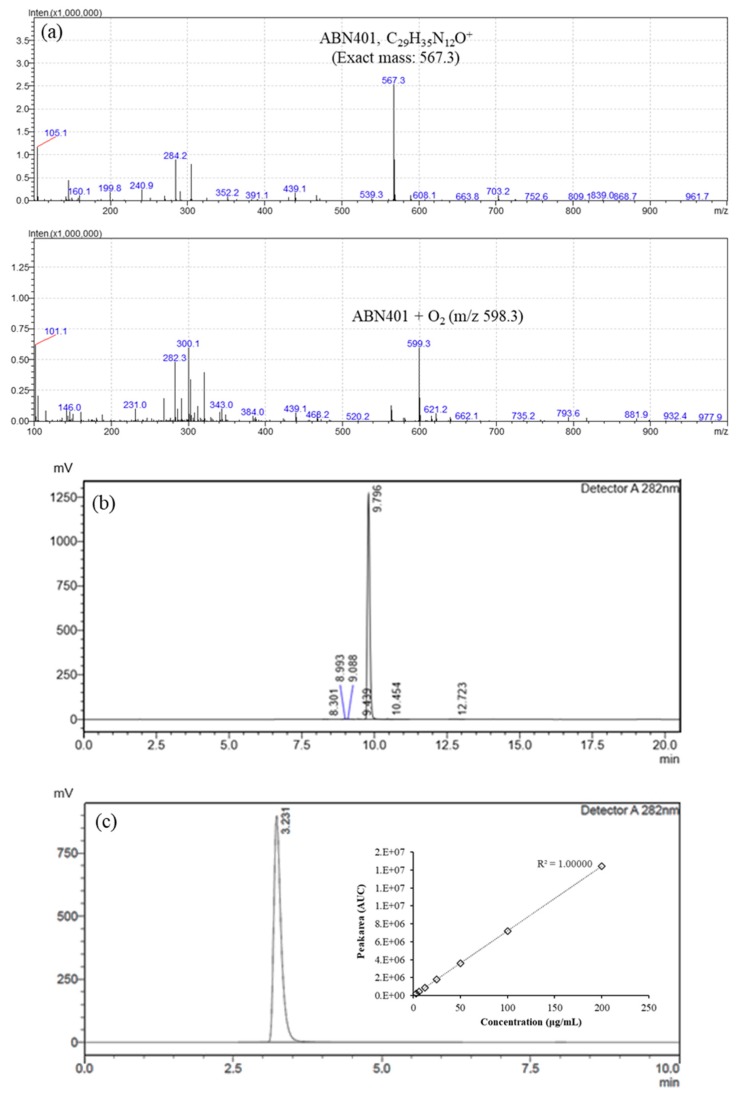
Determination of molecular weight of (**a**) ABN401 and addition of hydrogen peroxide using LC-MS, and HPLC result for ABN401 using (**b**) a qualification method (seven peaks) and (**c**) a quantification method (one peak) with linearity curve (R^2^ = 1) depending on concentration.

**Figure 3 pharmaceutics-12-00121-f003:**
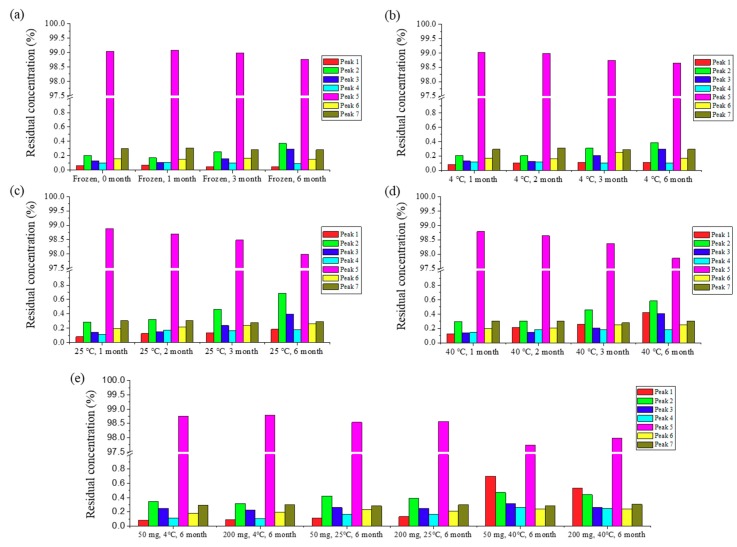
Six-month storage stability test for ABN401 (drug substance) at four different temperatures: (**a**) frozen, (**b**) 4 °C, (**c**) 25 °C, and (**d**) 40 °C in a closed high-density polyethylene (HDPE) container. HPLC was used to observe the behavior of the seven peaks depending on the time and temperature. In addition, (**e**) six-month storage stability test of granulated blend in capsule at two doses (50 mg and 200 mg) at different temperatures (4 °C, 25 °C, and 40 °C).

**Figure 4 pharmaceutics-12-00121-f004:**
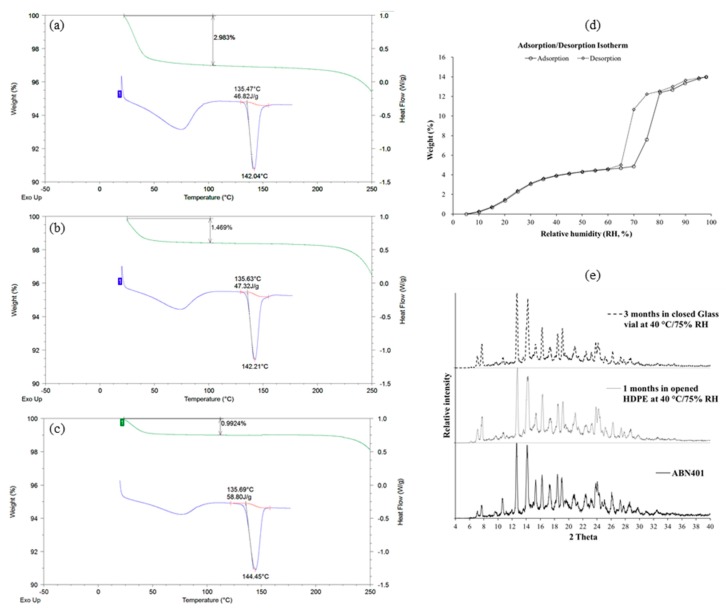
Thermal properties (TGA, green line (top) and DSC, blue line (bottom)) of ABN401 (**a**) from the first batch and (**b**) stored in open HDPE containers for 1 month at 40 °C, and (**c**) from second batch with relatively higher purity and less water content. (**d**) Dynamic vapor sorption (DVS) trends for ABN401 from second batch. (**e**) Powder X-ray diffraction (PXRD) results for ABN401 from the second batch (bottom) and from the first batch at two different conditions.

**Figure 5 pharmaceutics-12-00121-f005:**
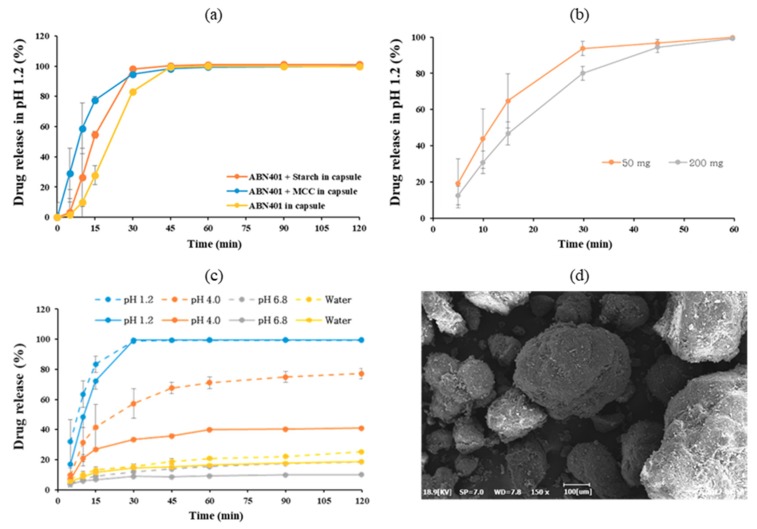
Dissolution profile of ABN401 in pH 1.2 of (**a**) physical blend in capsule with and without bulking agent (starch and MCC) at 50 mg, (**b**) a granulated blend in capsule at two doses (50 mg and 200 mg). (**c**) Dissolution profile of ABN401 at various pH values in gelatin capsule with optimized granulation at 50 mg (dotted) and 200 mg (solid) dose and (**d**) SEM image of the granulated blend.

**Figure 6 pharmaceutics-12-00121-f006:**
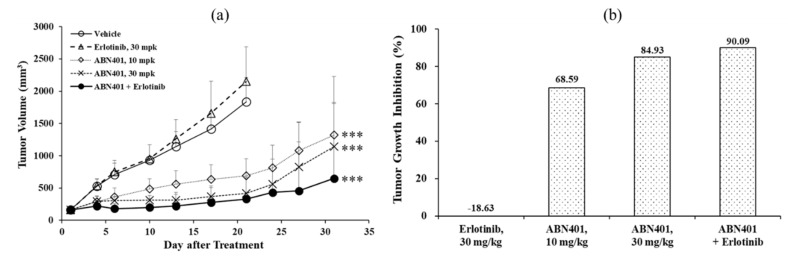
In vivo therapeutic efficacy of ABN401 in NSCLC patient-derived xenograft (PDX) models. (**a**) Growth curves of LU0858 NSCLC tumors in ABN401 and/or erlotinib-treated mice. Dose-dependent anti-tumor activity of ABN401 and in vivo anti-tumor activity of erlotinib alone or in combination with ABN401 is shown. *P*-values: *** < 0.001. (**b**) Tumor growth inhibition values on day 21 are shown.

**Table 1 pharmaceutics-12-00121-t001:** log *P* calculation (log K_ow_ (octanol/water) is regarded as log *D*).

Runs	pH	log *D*	log *P*
1	6.41	1.30	2.41
2	6.59	1.48	2.43
3	6.40	1.37	2.50
4	6.49	1.47	2.51
5	6.53	1.45	2.45
Average ± SD		1.41 ± 0.08	2.46 ± 0.04

**Table 2 pharmaceutics-12-00121-t002:** Equilibrium solubility measured at different pHs and excipients by HPLC.

Medium	Solubility (mg/mL)	±SD
Hydrochloric (pH 2.5)	25.97	0.920
Hydrochloric (pH 5.0)	3.92	0.090
1× PBS (pH 7.4)	0.01	0.0001
0.1 M acetate buffer (pH 4.0)	7.14	0.040
0.1 M acetate buffer (pH 4.5)	2.29	0.020
0.1 M acetate buffer (pH 5.0)	0.56	0.010
0.1 M citrate buffer (pH 4.0)	0.06	0.001
0.1 M citrate buffer (pH 4.5)	0.03	0.001
0.1 M citrate buffer (pH 5.0)	0.02	0.001
PEG 400 (20 %)	8.01	0.330
PEG 400 (40 %)	8.80	0.140
PEG 400 (60 %)	9.48	0.430
PEG 400 (80 %)	12.55	1.800
PEG 400 (100 %)	15.49	0.380
Cremophor EL (100 %)	14.12	1.810
Polysorbate 80 (100 %)	8.46	1.190

**Table 3 pharmaceutics-12-00121-t003:** ABN401 remaining in oral gavage formulation after 8 days of incubation.

Incubation in Cold Chamber at 2–8 °C				
Concentration (mg/mL)	pH	Day	Recovery (%)	SD
0.03	4.0	0	100.0	0.4
133		0	98.0	0.4
0.03	4.0	8	97.0	0.4
133		8	98.0	0.4
0.03	3.0	8	96.5	1.0
133		8	94.4	2.2
0.03	2.0	8	104.4	0.6
133		8	94.8	2.1

**Table 4 pharmaceutics-12-00121-t004:** Plasma pharmacokinetic parameters of ABN401 in beagle dogs following intravenous (IV) and oral (PO) administration of 1, 5, and 10 mg/kg body weight (mean ± SD).

Dose (mg/kg)	T_1/2_ (h)	C_max_ (ng/mL)	T_max_ (h)	AUC_last_ (h.ng/mL)	AUC_inf_ (h.ng/mL)	MRT_last_ (h)	BA (%)
IV, 1	1.9 ± 0.7	308.1 ± 270.8	0.083	266.3 ± 104.7	286.8 ± 107.7	1.5 ± 0.7	NA
PO, 1	9.4 ± 3.2	8.0 ± 3.8	1.3 ± 0.6	75.4 ± 42.0	96.2 ± 40.7	7.6 ± 2.5	28.3 ± 15.8
PO, 5	7.6 ± 4.5	50.8 ± 12.4	1.5 ± 0.7	408.7 ± 23.0	455.8 ± 28.5	8.6 ± 2.7	30.7 ± 1.7
PO, 5 *	4.9 ± 0.9	60.3 ± 29.1	2.0 ± 0.0	408.8 ± 201.3	425.4 ± 217.8	6.4 ± 0.7	30.7 ± 15.1
PO, 10	7.7 ± 5.8	52.5 ± 13.5	2.0 ± 0.0	302.2 ± 72.0	424.4 ± 186.2	6.3 ± 3.4	11.3 ± 2.7

* without microcrystalline cellulose (MCC), NA: not applicable

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
