# Peer review of "New Preclinical Development of a c-Met Inhibitor and Its Combined Anti-Tumor Effect in c-Met-Amplified NSCLC"

_pharmaceutics, 2020, doi:10.3390/pharmaceutics12020121_

Round 1
Reviewer 1 Report
Summary:
The authors present a series of physiochemical and pharmacokinetic studies for c-Met inhibitor ABN401.
Major comments:
The authors claim that 2nd generation c-Met inhibitors often fail due to renal toxicity issues associated with metabolites produced by aldehyde oxidase (AO). The authors presented in vitro experiments that suggest AO does not metalize ABN401, but not PK/metabolic studies that validate this in vivo. The authors should clearly address why this was not validated in vivo since some PK studies were performed in beagles. The renal toxicity most likely is dose-dependent. Would the doses suggested in the tumor efficacy (~30 mg/kg) study be considered safe? Would there be any notable interactions in the combination ABN401+erlotinib treatment ?Minor comments:
Page1: Line 21-22 (abstract): This sentence is vague because the structure. Suggested rewrite:
"Unlike other c-Met inhibitors that fail clinically, ABN401 is a newly synthesized c-Met inhibitor that is not degraded by aldehyde oxidase."
Page 2: Line 50-51 (intro): This sentence also needs better wording.
"However, c-Met inhibitors are still under investigation with on-going clinical trials for ~15 compounds."
In addition to experimentally determined physiochemical properties, it would also be beneficial to include calculated structure-based parameters such as polar surface area (PSA), H bond donor #, H bond acceptor #, and rotatable bond count. These parameters are often examined for structure-based profiling (Verber and Lipinski) to predict oral bioavailability and other PK parameters.
Author Response
Reviewer #1:
Major comments:
The authors claim that 2nd generation c-Met inhibitors often fail due to renal toxicity issues associated with metabolites produced by aldehyde oxidase (AO). The authors presented in vitro experiments that suggest AO does not metabolize ABN401, but not PK/metabolic studies that validate this in vivo. The authors should clearly address why this was not validated in vivo since some PK studies were performed in beagles. The renal toxicity most likely is dose-dependent. Would the doses suggested in the tumor efficacy (~30 mg/kg) study be considered safe? Would there be any notable interactions in the combination ABN401+erlotinib treatment?
The authors appreciate the reviewer’s comments and recommendations that the reviewer provided for the manuscript. The reviewer’s input has been invaluable to the authors during the revision process. For the clarity on AO metabolism study, following information is now updated in the manuscript and highlighted in yellow from line 449 to 456.
“The AO metabolism has several points that make it difficult to analyze in preclinical phase. First, standard metabolic stability studies using liver microsomes may be inaccurate since AO are mainly present in the cytosolic fraction. Second point would be variability of AO activities during storage conditions. Third, in vivo studies on AO-mediated metabolism in animal models are problematic because of differences in the profound species [16, 20-21]. Therefore, AO metabolism was only performed in human liver cytosol in the absence and presence of the AO inhibitor and a newly synthesized c-Met inhibitor, ABN401, was potentially not a substrate of AO.”
The purpose of PK in beagle dogs in the study is to investigate the adsorption and bioavailability of ANB401 at different doses and formulations in non-rodent species. Separate in vivo studies based on toxicology using rat/dog as well as metabolic stability using isotypes of CYPs were also performed. However, it was not presented in the manuscript since it’s considered to be out of scope in the paper but as a separate paper. The authors hope the reviewer consider this point.
In order to confirm the safety, a single dose acute toxicity test was performed as in-house test before the examination. 500 mg/kg and 1,000 mg/kg were administrated on male/female mice (n=3) including vehicle and observed for the next 14 days. As a result, no deaths occurred, no significant changes on major organs observed after necroscopy, and no notable change in body weight as well. Combination index (CI) of ABN401 with erlotinib was evaluated using H1373 and H1993 (i.e. NSCLC cell lines), and exhibited significant change in cell viability – with and without erlotinib. The related resulted is included in a separate paper and it is under review (i.e. Manuscript ID: cancers-647650, Therapeutic efficacy of ABN401, a highly potent and selective MET inhibitor, based on diagnostic biomarker test in MET-addicted cancer). The authors hope the information fulfils the reviewer’s thoughts.
Minor comments:
Page1: Line 21-22 (abstract): This sentence is vague because the structure. Suggested rewrite:
"Unlike other c-Met inhibitors that fail clinically, ABN401 is a newly synthesized c-Met inhibitor that is not degraded by aldehyde oxidase."
The authors appreciate the comments. The sentence is rewritten as following and highlighted in yellow (line 21-23). “Unlike other c-Met inhibitors that fail clinically, ABN401 is a newly synthesized c-Met inhibitor that is not potentially degraded by aldehyde oxidase in human liver cytosol.”
Page 2: Line 50-51 (intro): This sentence also needs better wording.
"However, c-Met inhibitors are still under investigation with on-going clinical trials for ~15 compounds."
The authors appreciate the consideration. The sentence is rewrote as follow and highlighted in yellow (line 50-52). “However, next generations of c-Met inhibitors are still under investigation with on-going clinical trials of 15 compounds.”
In addition to experimentally determined physiochemical properties, it would also be beneficial to include calculated structure-based parameters such as polar surface area (PSA), H bond donor #, H bond acceptor #, and rotatable bond count. These parameters are often examined for structure-based profiling (Verber and Lipinski) to predict oral bioavailability and other PK parameters.
The authors appreciate the reviewer keen points. Prior to determine physicochemical properties experimentally, the authors calculated the predicted parameters, but the log P (in silico: 0.54 ± 1.27, in vitro: 2.46 ± 0.04) was not the consistent between simulation and experiments, suggesting certain errors in predicting oral bioavailability and the other PK parameters for ABN401. However, the calculated surface area (PSA), H bond donor #, H bond acceptor #, and rotatable bond count is now updated in materials section (Line from 74 to 76). “Using an online server (http://admet.scbdd.com), values for polar surface area, number of hydrogen bond donor, number of hydrogen bond acceptor, and rotatable bond count were calculated as 119.04, 0, 13, and 7, respectively.”
Reviewer 2 Report
Review C-Met inhibitor
The article describes the preclinical development of ABN401 which is a c-met tyrosine kinase inhibitor.
The key hypothesis driving the work is that removing aldehyde oxidase activity will produce a better clinical candidate which does not produce insoluble metabolites.
The work is interesting and important but is missing the correct controls for the aldehyde oxidase assays which is essential to validate the authors hypothesis.
Major
In the section on metabolism (Fig 1b) the aldehyde oxidase control used is pthalazine but the control should be the known c-met inhibitors JNJ-38877605 or SGX523 otherwise the authors have no basis for their assumption that this assay is suitable for prediction of metabolism problems in c-met inhibitors. This is a major shortfall.
The in vivo study is interesting but no drug levels are reported so no pharmacodynamic conclusions can be drawn (L421).
Minor points
L36 and L38 use of word various – imprecise language consider giving examples to support the point being made.
L54 “other kinases” are mentioned but which ones? This is important for the reader to understand the deficiencies of current c-Met inhibitors.
L58 The structures or proposed structures of the insoluble metabolites should be shown.
L61 “It has a triazolopyridazine small….” Should be “It is a triazolopyridazine small…”
L71 materials section. Lacks clarity particularly last sentence.
L74 how was the ABN401 solubulised for the assays? No details are provided.
Figures.
Figure 1b pthalazine and menadione structures should be shown for clarity.
Author Response
Reviewer #2:
The article describes the preclinical development of ABN401 which is a c-met tyrosine kinase inhibitor. The key hypothesis driving the work is that removing aldehyde oxidase activity will produce a better clinical candidate which does not produce insoluble metabolites. The work is interesting and important but is missing the correct controls for the aldehyde oxidase assays which is essential to validate the authors hypothesis.
Major comments
In the section on metabolism (Fig 1b) the aldehyde oxidase control used is pthalazine but the control should be the known c-met inhibitors. JNJ-38877605 or SGX523 otherwise the authors have no basis for their assumption that this assay is suitable for prediction of metabolism problems in c-met inhibitors. This is a major shortfall. The in vivo study is interesting but no drug levels are reported so no pharmacodynamic conclusions can be drawn (L421).
The authors appreciate the reviewer’s keen comments and recommendations that the reviewer provided for the manuscript. The reviewer’s input has been invaluable to the authors during the revision process. The enzyme kinetics of aldehyde oxidase-catalyzed conversion of phthalazine to 1-phthalazinoe in pooled human liver cytosol is well known and often used in experimental technique at preclinical phases to observe a drug that can be potentially metabolized or inhibit human AO activity and to compare with menadione [1]. However, we believe that wordings were inappropriate in certain sentences mentioning “it is not degraded by AO” and made the reviewer uncomfortable. The expressions are now changed ‘potentially not degraded by AO’ (changed in line 22, 275, 456, and 500). ABN401 is been demonstrated to be potentially not a human aldehyde oxidase substrate in vitro. It remains to be clarified at clinical significance in vivo but the data will prove supportive data in analyzing clinical investigation. In addition, the present work is aimed to investigate physico-chemical properties of the compound and developing two independent formulations for PDX study as well as feasibility on non-rodent species.
On the other hand, in order to confirm the safety with dose levels, a single dose acute toxicity test was performed in house before the examination. 500 mg/kg and 1,000 mg/kg were administrated on male/female mice (n=3) including vehicle and observed for the next 14 days. As a result, no deaths, no significant changes on major organs after necroscopy, and no notable change in body weight as well. IC50 and Combination index (CI) of ABN401 with erlotinib was evaluated using H1373 and H1993 (i.e. NSCLC cell lines), and exhibited significant change in cell viability – with and without erlotinib. The related resulted is included in a separate paper and it is under review (i.e. Manuscript ID: cancers-647650, Therapeutic efficacy of ABN401, a highly potent and selective MET inhibitor, based on diagnostic biomarker test in MET-addicted cancer). The authors hope the information fulfils the reviewer’s thoughts.
[1] Obach, R.S.; Huynh, P.; Allen, M.C.; Beedham, C. Human liver aldehyde oxidase: inhibition by 239 drugs. J Clin Pharmacol 2004, 44, 7-19.
Minor points
L36 and L38 use of word various – imprecise language consider giving examples to support the point being made.
The authors appreciate the reviewer’s comment. The word ‘various’ is now deleted and corrections are made. The update is highlighted in yellow from line 38 to line 41.
L54 “other kinases” are mentioned but which ones? This is important for the reader to understand the deficiencies of current c-Met inhibitors.
The authors appreciate the comment. Wrongly inhibited targets by first generations of c-Met inhibitors is now updated and highlighted in yellow from line 54 to 55.
L58 The structures or proposed structures of the insoluble metabolites should be shown.
The authors appreciate the consideration. The chemical structure of the metabolites is now updated with better wordings and explanation. It is highlighted in yellow from line 59 to 62.
L61 “It has a triazolopyridazine small….” Should be “It is a triazolopyridazine small…”
The authors appreciate the comments. The correction is done and is highlighted in yellow (line 64).
L71 materials section. Lacks clarity particularly last sentence.
The authors appreciate the comments. The last sentence is corrected and highlighted in yellow (line 77).
L74 how was the ABN401 solubulised for the assays? No details are provided.
The authors appreciate the comment. It was solubilized in 0.3% DMSO. The correction is done and is highlighted in yellow (line 80).
Figures.
Figure 1b pthalazine and menadione structures should be shown for clarity.
The authors appreciate the comment. The structures of phthalazine and menadione is now added in Figure 1 b.
Round 2
Reviewer 2 Report
The authors have made a reasonable attempt to revise the manuscript. Its a shame they don't have the control data on the AO assay but I guess that this might involve a lot of extra experiments.